# Therapeutic Efficacy of Anti-Bestrophin Antibodies against Experimental Filariasis: Immunological, Immune-Informatics and Immune Simulation Investigations

**DOI:** 10.3390/antib10020014

**Published:** 2021-04-17

**Authors:** Nabarun Chandra Das, Anindya Sundar Ray, Jagadeesh Bayry, Suprabhat Mukherjeee

**Affiliations:** 1Integrative Biochemistry & Immunology Laboratory, Department of Animal Science, Kazi Nazrul University, Asansol 713340, West Bengal, India; nabarunchandradas@gmail.com (N.C.D.); argharay90@gmail.com (A.S.R.); 2Institut National de la Santé et de la Recherche Médicale, Centre de Recherche des Cordeliers, Sorbonne Universités, Université de Paris, F-75006 Paris, France; 3Indian Institute of Technology Palakkad, Palakkad 678623, Kerala, India

**Keywords:** lymphatic filariasis, filarial parasites, bestrophin, anti-FSAg antibody, antibody therapy, in silico analyses

## Abstract

Lymphatic filariasis (LF) is a debilitating parasitic disease caused by filarial parasites and it is prevalent across the underprivileged population throughout the globe. The inadequate efficacy of the existing treatment options has provoked the conception of alternative strategies, among which immunotherapy is steadily emerging as a promising option. Herein, we demonstrate the efficacy of an antibody-based immunotherapeutic approach in an experimental model of filariasis, i.e., Wistar rat infected with *Setaria cervi* (a model filarial parasite). The polyclonal antibodies were raised against filarial surface antigen bestrophin protein (FSAg) in mice using the purified *Wuchereria bancrofti* FSAg. The adoptive transfer of anti-FSAg antibody-containing serum resulted in the significant reduction of parasite burden in filaria-infected rats. Intriguingly, anti-FSAg sera-treated animals also displayed a reduction in the level of proinflammatory cytokines as compared to the infected but untreated group. Furthermore, our in silico immunoinformatics data revealed eight B-cell epitopes and several T-cell epitopes in FSAg and these epitopes were linked to form a refined antigen in silico. The immune simulation suggested IgM and IgG1 as the predominant immunoglobulins induced in response to FSAg. Taken together, our experimental and simulation data collectively indicated a therapeutic potential of anti-FSAg sera against LF.

## 1. Introduction

Lymphatic filariasis (LF) is a vector-borne parasitic disease of human caused by the pathogenic filarial nematodes, viz., *Wuchereria bancrofti*, *Brugia malayi* and *Brugia timori*, that predominantly affects the lymphatic system to cause permanent disabilities and associated morbidities [1,2]. This disease is responsible for social stigma and huge economic loss in the places considered endemic for this disease. Currently, 49 countries with an estimated population of 120 million suffer from this disease while 893 million are at risk of infection [2]. It is noteworthy to mention that India is one of the biggest contributors (~40%) of global burden of LF. World Health Organization (WHO) has categorized this disease under ‘Neglected Tropical Diseases’, since LF is a problem of developing countries, and specifically underprivileged populations are struggling with this disease. WHO started a global program to eradicate LF by 2020, which seems to have been an unsuccessful attempt, and now this program has been re-structured to achieve the target within 2030 [3]. This is due to the limited efficacy of the available anti-filarial chemotherapeutics (albendazole, ivermectin, diethylcarbamazine), the emergence of resistance against the drugs, and adverse side-effects [4].

Considering the current scenario, conception of immunotherapeutic intervention approaches is steadily emerging as an alternative option. Disruption of the immune-homeostasis of the host is considered to be one of the key strategies adopted by the filarids. On the other hand, uncontrolled inflammatory events contribute to the overt immunopathology of LF that majorly includes lymphedema, lymphangitis and elephantiasis [5]. Such inflammatory events are known to be induced by the host–parasite interactions, wherein surface-antigens of the parasite interact with the innate immune receptors (e.g., Toll-like receptor 4, TLR4) resulting in the production of proinflammatory cytokines and chemokines from the immune cells [6,7].

In this context, a novel surface protein, namely FSAg/MfP, belonging to the nematode-bestrophin superfamily, was reported to be one of the crucial inducers of inflammatory consequences associated with bancroftian filariasis [7]. This protein is present in the cuticle of the filarid across all the developmental stages of the parasite [8]. FSAg acts a ligand for TLR4 located on the surface of human antigen presenting cells like macrophages and dendritic cells [6,7]. Binding of FSAg to macrophage-TLR4 resulted in the M2-M1 polarization through the activation of NF-κB pathway and subsequent expression of the proinflammatory mediators [7]. Interaction of FSAg with dendritic cell TLR4 induced maturation as well as activation of these antigen presenting cells [6]. Intriguingly, FSAg-educated dendritic cells were found to trigger polarization of naïve T-cells towards Th1 and regulatory T cell subtypes [6]. Considering the strong influence of FSAg in the immunopathogenesis of LF, this protein could be the target for immunotherapeutic intervention of this chronic inflammatory disease.

Based on this postulate, we raised polyclonal antibodies against FSAg and examined their therapeutic efficiency when used in the form of a serum against filariasis in an experimental model. Experimental filariasis was induced by *Setaria cervi*, a WHO recommended model filarial parasite that has been extensively used in filarial research due its similarity with the *Wuchereria bancrofti* at biochemical and immunological level. Our experimental and immune simulation data collectively indicated a therapeutic potential of anti-FSAg sera against LF.

## 2. Materials and Methods

### 2.1. Ethical Clearance

The study was approved by the institutional ethical committee of Kazi Nazrul University, Asansol-731340, West Bengal. The investigation involving human blood samplings was conducted following the rules of the Declaration of Helsinki of 1975 (https://www.wma.net/what-we-do/medical-ethics/declaration-of-helsinki/ (accessed on 8 June 2020)).

### 2.2. Antibodies and Consumables

All the reagents used in the study were of analytical grade and were commercially available in India. ELISA kits were obtained from Ray Biotech (Kolkata, India). Phosphate buffer saline and antibiotic cocktail were procured from HI Media (Kolkata, India). Disposal syringes and glass slides were obtained from Dispo Van (Kolkata, India) and Riviera (Kolkata, India), respectively. Giemsa’s stain, methyl alcohol and chloroforms were purchased from Merck (Kolkata, India).

### 2.3. Parasite

Microfilariae of *W. bancrofti* were obtained from the filaria-endemic villages of Bankura, West Bengal following the procedure described in Mukherjee et al., 2017 [7]. On the other hand, the model filarial parasite *S. cervi* was obtained from the local abattoirs located in Asansol, West Bengal as per the protocol of Mukherjee et al., 2020 [8]. The adult female parasites were dissected and microfilariae were obtained [8].

### 2.4. Development of Antibody

Anti-FSAg antibody was raised following the procedure detailed in Mukherjee et al., 2017 and 2019 [6,7]. In brief, FSAg antigen was isolated from the surface of *W. bancrofti* and validated as previously described by us [6,7]. The antigen in 1X PBS (100 µg/mL) was mixed with the adjuvant and injected to adult BALB/c mice (*n* = 10) through the foot-pad route. The first dose was administered with Freund’s complete adjuvant, while the second dose was given with Freund’s incomplete adjuvant after 7 days. The control group include mice injected with Freund’s complete adjuvant mixed with PBS (adjuvant control). After three weeks from the last dose, sera were collected from the blood samples. Anti-FSAg antibody titers in the sera were determined by ELISA as previously described [7] using purified FSAg.

### 2.5. Experimental Infection Model of Filariasis

The experimental filariasis model was developed using Wister albino rats following Mukherjee et al., 2017 [9]. Live and motile adult female *S*. *cervi* were dissected by cutting the mouth and tail part. Microfilariae (Mf) were treated with antibiotic cocktail (100 U/mL penicillin, 100 μg/mL streptomycin, 0.25 μg/mL amphotericin B and 10 µg/mL polymyxin B) for 1 h, washed with 1X PBS and Mf (*n* = 3.175 × 10^5^) were aseptically transplanted intraperitoneally into anesthetized adult male Wister rats (*n* = 12; 120 ± 10 g body weight). All the rats were diagnosed for infection at every 5 day interval up to 120 days post infection. Blood samples were collected from each experimental animal through tail clipping. Blood smear was prepared over glass slide, air dried and de-hemoglobinized. Slides were fixed with absolute methyl alcohol, dried and stained with Giemsa for 7 min. Slides were washed with tap water, dried in air and observed under optical microscope (Olympus, India) and Mf were counted manually.

### 2.6. Treatment Schedule

The therapeutic efficacy of anti-FSAg polyclonal antibodies in the form of a hyperimmune sera was examined in filaria-infected rats. The highest dilution of the serum at which the antigen-antibody interaction was prominently detected in an ELISA was considered as the antibody titer (endpoint). The therapeutic doses of antibodies were prepared by adjusting the volume of serum. The infected rats were divided into six groups (*n* = 5) and were treated with different doses of anti-FSAg antibody containing serum (10, 20, 50, 100, and 200 µL/100 gm body weight) through intraperitoneal injection with sterile PBS. An infected untreated set was included as a control. A healthy normal set of animals was also included. Hyperimmune serum was injected on the 10th day post infection and thereafter examined for another 10 days. After 10 days, all the animals were examined for parasite load through tail clipping followed by Giemsa staining of blood smears. Inflammatory cytokines in the circulation were determined by ELISA.

As a positive control, animals were treated with a single dose of diethylcarbamazine at 200 mg/kg of body weight. The treated animals were kept for 10 days. After 10 days, levels of inflammatory cytokines in the circulation were determined by ELISA.

In another set of experiments, we also examined the prophylactic effect of anti-FSAg sera against *S. cervi* infection in rats. Six group of rats (*n* = 5) were treated with two different doses of anti-FSAg serum, i.e., 100 and 200 µL/100 gm body weight) through intraperitoneal injection and after 10 days, rats were experimentally infected with *S. cervi* as depicted in the previous section. On the 10th day post infection, *S. cervi* infection was monitored through blood smear followed by Giemsa’s staining.

### 2.7. Determination of the Inflammatory Parameters

We collected blood samples from the antibody-treated and untreated animals, and centrifuged to obtain the serum. Serum levels of TNF-α and IL-1β were estimated by sandwich ELISA using the ELISA kits provided by Ray Biotech.

### 2.8. Statistical Analysis

All experiments were performed in triplicate sets with at least five replicates. Data are presented as mean ± S.D. and differences among the data were determined by One-way ANOVA using Graphpad Prism 8.0. Statistical significance is indicated in the figure legends.

### 2.9. Immuno-Informatics and Immune Simulation Studies

In addition to the experimental studies, we have extended our work to bio-computational approaches to investigate the immunogenicity of FSAg, development of a vaccine construct using the epitopes of FSAg and in silico prediction of the immune response. Before designing the experiment, we retrieve the amino acid sequence of *W. bancrofti* FSAg (Accession number: YCEL_CAEEL) from Swiss Prot database (www.uniprot.org/uniprot/ (accessed on 6 December 2020)) in FASTA format and further experiments were conducted subsequently.

### 2.10. Prediction of Linear B-Cell Epitopes

Prediction of the presence of linear B-cell epitope in a given antigen is one of the crucial steps in analyzing immunization and antibody production *in silico* [10]. Herein, ABCpred web server (http://crdd.osdd.net/raghava/abcpred/ (accessed on 6 December 2020)) that runs on recurrent neural network technique was primarily used to identify potential B-cell epitopes in FSAg [11]. Furthermore, BCPREDS (http://ailab-projects1.ist.psu.edu:8080/bcpred/ (accessed on 6 December 2020)), SVMTriP (http://sysbio.unl.edu/SVMTriP/ (accessed on 6 December 2020)) and Immune Epitope Database (IEDB) (https://www.iedb.org/ (accessed on 6 December 2020)) servers were also used to verify and screen actual B-cell epitopes [12,13,14].

### 2.11. Prediction of T-Cell Epitopes

Beside B-cell epitopes, prediction of T-cell epitopes is also important in predicting immune responses as an antigen having T-cell epitopes can efficiently activate CD4 T-cells to shape adaptive immunity [15,16]. To identify the MHC II epitopes from FSAg, ProPred (http://crdd.osdd.net/raghava/propred/ (accessed on 6 December 2020)) servers were explored respectively [17,18]. ProPred server uses matrices prediction algorithm method and predict epitopes on the basis of recognition of 51 MHC-II allele.

### 2.12. Prediction of Antigenicity

Prediction of antigenicity is an important aspect in determining the potential epitopes present in an antigen and for designing a refined antigen construct. VaxiJen v2.0 server was used to determine all possible epitopes in FSAg through an alignment-independent prediction method [19].

### 2.13. Design of Multi-Epitope Construct

Considering the efficacy of anti-FSAg antigen in reducing parasite load and ameliorating filarid-induced inflammation as well as inspired from immunoinformatic clues suggesting occurrence of B- and T cell epitope, a multi-epitope construct was prepared using in silico. T cell epitopes were predicted through in silico approach from FSAg-derived B-cell epitopes and linked together with GPGPG and AAY linkers, to prepare the antigen.

### 2.14. Immune Simulation

Immune simulation is one of the easiest ways to evaluate the immunogenicity and analyze immune responses in an in silico environment. Herein, C-immSim server (http://150.146.2.1/C-IMMSIM/ (accessed on 8 December 2020)) was used to predict the molecular insights between the immunogenic molecules at mesoscopic level [20]. This server works using a machine learning method and uses the FASTA format of designed epitope construct as an injection. The algorithm was the default and we tested the efficacy of the refined FSAg using the default parameters applicable for studying human immune response. In this report, to study the threshold immune response, we administered three injections at 1, 84 and 168 timestep intervals. In real life, each timestep indicates 8 h and each interval is about 28 days.

## 3. Results

### 3.1. Development of Filarial Infection in Rat Model

In the present study we developed an animal (rat) model of LF using WHO recommended filarial parasite *S. cervi*. Initially, we administered a variable number of *S. cervi* larvae (Mf) to rats through intraperitoneal route and found that 3.0 × 10^5^ Mf could induce stable infection with least mortality on the 10th day post infection (data not shown). After optimizing the inoculum size, we performed experimental infection in rat and found that 1/6th of the initially injected Mf were present in the blood stage (patent) infection (Figure 1). This infection was maintained for up to 70 days with a gradual decrease in Mf load, and after that a significant (*p* < 0.05) decrease was noted for up to 120 days (Figure 1).

### 3.2. Anti-Bestrophin Antibody Reduces Microfilarial Load

We raised anti-FSAg polyclonal antibodies in BALB/c mice. These anti-FSAg polyclonal antibodies displayed good specificity to the target antigen, as shown in Figure 2A. The antibody titers in the sera of FSAg-immunized mice were higher than that of control adjuvant-treated animals (Figure 2A). Sera from control animals showed only marginal levels of reactivity to FSAg. We did not observe either external allergic changes or alterations in the morphology of internal organs of immunized animals. Our present data corroborate the earlier report on the efficiency of FSAg in inducing high titered antibody response in the mice model [7].

We treated *S. cervi*-infected rats with different doses of anti-FSAg sera and found that anti-FSAg sera treatment caused a reduction in the parasite load 10 days after first injection (Figure 2B,C). Anti-FSAg sera at doses of 100 µL/100 gm body weight or above were found to be highly effective in clearing the parasite load. In fact, at this dose, almost all parasites were effectively removed from the rats (*p* < 0.0001, in comparison to the control group) (Figure 2B). Therefore, further experiments were performed using 100 µL/100 gm body weight dose. Lower doses of antibodies such as 10 µL and 20 µL/100 gm body weight were not effective (Figure 2B).

Furthermore, we also observed that anti-FSAg sera possessed a prophylactic effect and provided partial protection against experimental filarial infection in the rats (Figure 2D,E). Anti-FSAg sera at a dose of 100 µL/100 gm body weight was found to confer significant protection and lowered the infectivity of *S. cervi* in the rats. However, after comparing the two different approaches of administration, the efficacy of anti-FSAg antisera was found to be greater when applied in *S. cervi*-infected animals.

### 3.3. Anti-Bestrophin Antibody Reduces Microfilaria-Induced Inflammation

Microfilaria-induced inflammation is considered to be one of the key events in filarial immunopathogenesis, wherein FSAg has been reported as an important regulator of inflammation [6,7]. Therefore, we checked whether anti-FSAg sera could play a role in inhibiting the microfilaria-induced inflammation in infected rats. As depicted in Figure 3A,B, a high level of inflammation was noted in the filaria-infected animals as compared to the uninfected/control animals. In contrast, anti-FSAg sera at the dose of 100 µL/100 gm body weight significantly reduced the levels of two major proinflammatory cytokines, viz. TNF-α and IL-1β, in the circulation of microfilariae-infected animals. However, dose dependent effect of anti-FSAg sera was not observed in our experiments. The ability of anti-FSAg sera to reduce the levels of TNF-α and IL-1β was comparable to diethylcarbamazine-treated group (positive control) (Figure 3).

Previously, by using an acute treatment schedule, we showed that treatment of *W. bancrofti* microfilariae with anti-FSAg sera impairs the pro-inflammatory ability of the parasite [7]. Importantly, even under therapeutic settings, we found that anti-FSAg sera could significantly reduce the parasite load as well inflammatory cytokines in vivo (Figure 2 and Figure 3).

### 3.4. FSAg Possesses Both B-Cell and T-Cell Epitopes

Our experimental findings indicated an excellent immunogenicity of FSAg and long-term efficacy of anti-FSAg sera in reducing the parasite load, as well as inflammation in the experimental model of filariasis. To elucidate the immunobiological attributes of FSAg and its ability to induce antibody response in humans, we resorted to in silico approaches. Our in silico analyses revealed the presence of eight B-cell epitope peptides in the sequence of FSAg comprising 16–20 amino acids (Table 1). On the other hand, T-cell epitope prediction analysis demonstrated the occurrence of five MHC-II epitopes from the selected B-cell epitopes present in FSAg (Table 2).

We linked these epitopes using GPGPG and AAY linkers to ultimately design the refined antigen-construct (Table 3) for studying the immune response. While preparing the construct, linkers were used to provide flexibility, folding and stability in the structure of the refined antigen. Herein, a VaxiJen score of 0.5968 for the refined antigen primarily confirmed a high antigenicity of the protein.

### 3.5. Anti-FSAg Antibody-Mediated Immune Response Is Mediated by IgG and IgM

Through immune simulation, we examined the immune responses generated in response to repeated exposure to refined FSAg. Our simulation data revealed that FSAg induces high humoral immune response in the mammalian system (Figure 4). As shown in Figure 4A, refined FSAg induces weak primary immunoglobulin response after first immunogen exposure but second exposure demonstrated elevated immunoglobulin response with high IgM+IgG response. The major share of immune response during this stage was apparently mediated by IgM. Subsequent exposure to refined FSAg further raised IgM+IgG titers but at this stage, the intensity of IgM and IgG responses were similar. IgG response was mainly due to IgG1 while the contribution of IgG2 was negligible. Such a high IgM+IgG response was further supported by an amplified population of diverse B-cell subpopulations: memory B cells, and B cells expressing IgM and IgG1 isotypes (Figure 4B,C). The simulation assay also suggested persistence of active antibody-producing B cells for a prolonged period (Figure 4D).

## 4. Discussion

Lack of mechanistic knowledge of the filarial immunopathogenesis has been the major impediment in executing the WHO-initiated global filariasis eradication program. Impairment of immune homeostasis characterized by chronic inflammatory milieu is the major pathological hallmark of LF in human. As stated earlier, the limited efficacy of available chemotherapeutics and emergence of drug resistance have been the major concern in combating new cases of infection. Moreover, the ongoing COVID-19 pandemic has also come out as a new challenge in controlling several infectious diseases registered under ‘Neglected Tropical Diseases’ including LF [3]. Particularly, diagnosis of new asymptomatic LF cases as well as treatment and management of symptomatic patients has been immensely interrupted throughout the globe due to COVID-19 pandemic situation. As a result, WHO has proposed a new 2021–2030 road map to implement global filariasis eradication mission [3]. In this context, conception of an alternative therapy is of utmost necessity to counteract the myriad threat posed by the filarial parasites. Increasing evidence in the literature suggests that immunotherapy could be an option for easing the inflammatory consequences induced by the filarial parasite and to confine the immunopathological transformation of normal lymphatics to elephantiasis [6,21].

In this investigation, we present the in vivo efficacy of anti-bestrophin antibodies in treating the experimental filarial infection in the rat model. The adopted sera therapy not only reduced the parasite burden but also exerted anti-inflammatory effects by reducing the circulating levels of inflammatory cytokines. Previous studies conducted by Mukherjee et al. [7] demonstrated that a 70 kDa phosphorylcholine-binding novel antigen (namely FSAg) of the parasite *W. bancrofti* (the major causative parasite of filariasis in human) is responsible for inducing the inflammatory responses in the host by causing classical macrophage activation through TLR4-NF-κB signaling axis. Moreover, FSAg also induced maturation and activation of human dendritic cells and secretion of pro-inflammatory cytokines through TLR4-mediated activation of cells [6]. In addition, FSAg also directed dendritic cells to cause polarization of naïve T cells toward Th1 and regulatory T cell subsets [6]. Therefore, FSAg appears to be a meaningful target for developing immunotherapeutic strategies against lymphatic filariasis. In this direction, our previous finding suggested that antibodies against FSAg could selectively mask surface FSAg in the microfilaria and could alleviate filaria-induced inflammation [7]. Taking a clue from these observations, we aimed at exploring the therapeutic use anti-FSAg sera in filaria-infected animal model.

However, unavailability of an appropriate animal model for *W. bancrofti* is a major problem for filarial researchers. Herein, we selected *S. cervi* as a model filarial parasite that could be easily maintained in rats [22,23]. *S. cervi* possess significant similarity with *W. bancrofti* at the level of life cycle, pathogenesis, antigenicity, and metabolism as well as other physiological processes [24,25,26,27]. Intriguingly, a homolog of *W. bancrofti* FSAg has also been reported recently from *S. cervi* [8]. In addition, we have previously demonstrated that *W. bancrofti* FSAg and *S. cervi* FSAg share a high degree of antigenicity [8]. Herein, we performed experimental infection of rats with *S. cervi* and thereafter treated them with anti-FSAg sera. The sera (antibodies) treatment yielded promising results as both worm burden as well as inflammation were subdued with no notable side reactions. Moreover, prophylaxis use of anti-FSAg sera also exerted protective effects against *S. cervi* infection in the rats. However, the efficacy of anti-FSAg sera was found to be greater when applied therapeutically in *S. cervi*-infected animals.

Although anti-FSAg serum was raised in mice, it was compatible in the rat model. By adoptive transfer of anti-FSAg sera, we aimed at targeting the parasite through FSAg. Anti-FSAg sera though could be induced in rats, the amount of protein required for this purpose is higher and the level of response was low. On the other hand, high antibody titered anti-FSAg sera could be raised in mice with lesser protein antigen. Moreover, adoptive transfer of mice anti-FSAg sera did not trigger any perturbations in the cytokine milieu of rat. Production of large amounts of recombinant FSAg in the near future and selection of an appropriate adjuvant would solve these issues and allow us to obtain rat anti-FSAg sera for the experiments.

Furthermore, to determine whether FSAg could also induce immune response in humans, we employed immunoinformatics and immune simulation approaches [28] to obtain mechanistic insights on the humoral immune response induced by FSAg. We found that FSAg contains numerous B-cell epitopes, and moreover, these B cell epitopes contained several T-cell epitopes based on the recognition of 51 MHC-II alleles. These data suggested that an appropriate antigen could be constructed to induce high-titered antibody responses in the individuals and to reduce filarial burden in the patients. In fact, our immune simulation data predicted that IgM and IgG1 dominate the immune response generated in response to refined FSAg and this antigen induces long-living B cell response. Therefore, we suggest that immunotherapeutic approaches based on refined FSAg could be undertaken in the population for the long-term protection against filarial infection. Although we used Freund’s complete adjuvant in mice, we need to identify an ideal adjuvant that could be used for the immunization purpose in humans.

Antibodies have been used in the therapy of a wide range of infectious diseases including recent COVID-19 caused by SARS-CoV-2 [29,30,31]. Our experimental results collectively indicate that anti-FSAg-based antibody therapy could be a promising option to treat filarial infection and associated immunopathology. In particular, deigning a vaccine based on the epitopes of FSAg seems to be a target in near future. Moreover, this study will provide a platform to the filarial researchers to conduct additional antibody-directed therapies.

## 5. Conclusions

Our experimental studies indicated that antibodies raised against *W. bancrofti* bestrophin could reduce the worm burden as well as inflammatory consequences associated with LF. Immune simulation data indicated towards the possibility of designing effective anti-filarial vaccine (which is not available to date) candidate using the epitopes of FSAg. However, this claim needs additional experimental validation in non-human primates for further pre-clinical development.

## Figures and Tables

**Figure 1 antibodies-10-00014-f001:**
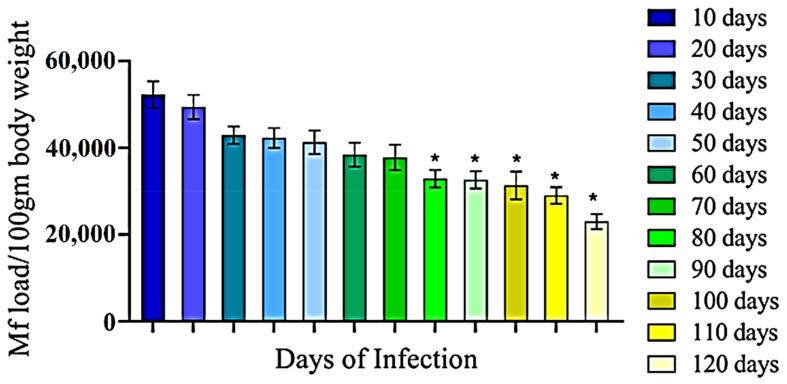
Optimization of filarial infection in rat model. Rats were experimentally infected with *S. cervi* microfilariae through intraperitoneal injection. Infection was diagnosed by tail-clipping followed by thick smear and Giemsa staining. All the experiments for each set were conducted in triplicate and repeated at least five times. Data are presented as mean ± S.D. Differences among the sets were compared with the 10 day parasite by One-way ANOVA. * *p* < 0.01.

**Figure 2 antibodies-10-00014-f002:**
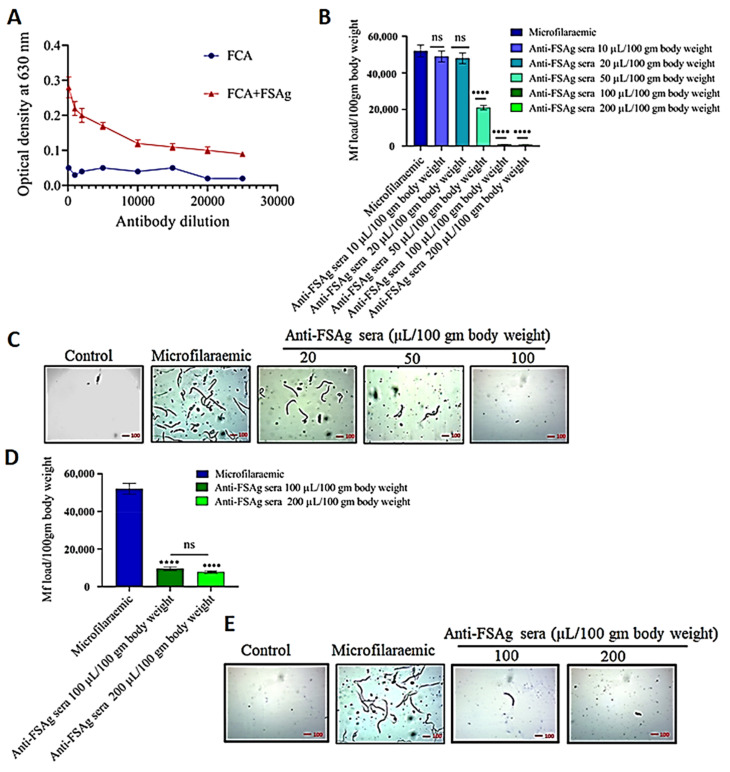
Induction of anti-bestrophin (FSAg) antibodies in BALB/c mice (**A**) and determination of their efficacy in an experimental filariasis model (**B**). Filaria-infected rats were treated with anti-FSAg sera with indicated doses and microfilarial load was determined by analyzing the blood smear following Giemsa staining (**C**). (**D**,**E**) Prophylactic effect of anti-FSAg sera in providing protection against *S. cervi* infection. Rats were pretreated with anti-FSAg sera and thereafter experimentally infected with *S. cervi.* Infection was monitored by tail clipping followed by Giemsa’s staining of blood smears. All the experiments were performed in triplicates and repeated five times. Statistical significance among the antibody treated groups were compared by One-way ANOVA. **** *p* < 0.0001; ns, not significant.

**Figure 3 antibodies-10-00014-f003:**
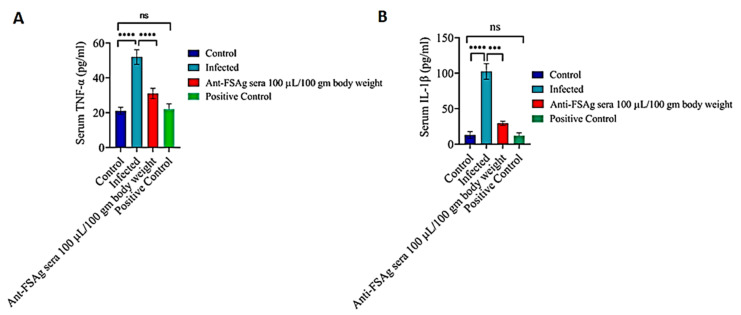
Anti-bestrophin (FSAg) antibody ameliorates microfilaria-induced proinflammatory responses. Anti-FSAg sera treatment caused reduction in the levels of serum TNF-α (**A**) and (**B**) IL-1β in filaria-infected rats. Levels of the cytokines were determined by sandwich ELISA. All the experiments were performed in triplicates and repeated for at least five times. Diethylcarbamazine (200 mg/kg) was used as a positive control. Difference among the data sets were compared by One-way ANOVA. *** *p* < 0.001, **** *p* < 0.0001.

**Figure 4 antibodies-10-00014-f004:**
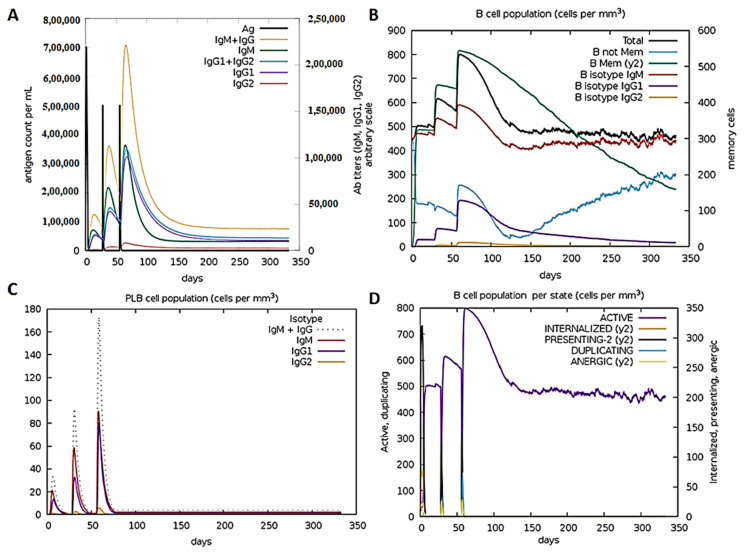
Prediction of humoral response to refined FSAg. (**A**) IgM and IgG1 principally govern the primary and secondary immune response. (**B**) Abundance of B cell subpopulations that produce antibodies to refined FSAg. (**C**,**D**) Comparison of different B cell populations induced in response to refined FSAg in immune simulation platform.

**Table 1 antibodies-10-00014-t001:** Predicted B-cell epitopes in W. bancrofti FSAg protein.

Sl. No.	Peptide Sequence	Start Position	End Position
1	KGSIWRSVWREL	17	28
2	YTDPTDSKGYRKIFKVMCNE	48	67
3	NVVSR	87	91
4	LLTEKEYEKLQDINEPSPG	159	177
5	ITAGRGSVNYVSVATN	195	210
6	RISF	216	219
7	QDDDPPKFEEDVFWKHHNEQQQMHQSMFLPRVPT	321	354
8	HPPKLHTYLEMKNQDPEEYKNRKNN	370	394

**Table 2 antibodies-10-00014-t002:** Computational prediction of MHC-II epitopes from FSAg-derived B-cell epitopes.

Sl. No.	B-Cell Epitope	MHC-II Epitope	VaxiJen Score
1	LLTEKEYEKLQDINEPSPG	LLTEKEYEK	1.1015
2	QDDDPPKFEEDVFWKHHNEQQQMHQSMFLPRVPT	WKHHNEQQQ	0.7647
		MHQSMFLPR	0.7518
3	HPPKLHTYLEMKNQDPEEYKNRKNN	MKNQDPEEY	0.8284
		LHTYLEMKN	0.9114

**Table 3 antibodies-10-00014-t003:** Formulation of epitope-based refined antigen-construct for FSAg.

Sequence	Length
KGSIWRSVWGPGPGLLTEKEYEKGPGPGQxQMHQSMFLGPGPG	223
PKFEEDVFWGPGPGKHHNEQQQMGPGPGDPPKFEEDVGPGPG	
PPKFEEDVFGPGPGPKLHTYLEMGPGPGDPEEYKNRKGPGPGP	
KLHTYLEMGPGPGKLHTYLEMKGPGPGEEYKNRKNNAAYLL	
TEKEYEKAAYWKHHNEQQQAAYMHQSMFLPRAAYMKNQDP	
EEYAAYLHTYLEMKN	

## Data Availability

Data will be available on request to S.M.

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
