# Peer review of "Therapeutic Efficacy of Anti-Bestrophin Antibodies against Experimental Filariasis: Immunological, Immune-Informatics and Immune Simulation Investigations"

_2073-4468, 2021, doi:10.3390/antib10020014_

Round 1
Reviewer 1 Report
Authors performed passive transfer of antisera against a microfilarial sheath antigen (FSAg) to rats infected with animal filaria Setaria cervi, to evaluate its efficacy of immunotherapy on filarial infection and filariasis. The results showed the significant reduction of microfilaria burden in blood and inhibition of pro-inflammatory cytokines that contribute to morbidity. Authors further predicted several B-cell and T-cell epitopes in FSAg using in-silico algorithm and predicted the construct from these epitopes may induce high titer of IgM and IgG1 in immunized host. The results provide further evidence FSAg could be a vaccine or immunotherapeutic target against LF. However, It means some questions and concerns needed to be addressed:
- How was FSAg protein produced? Is it a parasite-derived product or a recombinant protein? Is it from human filaria Wuchereria bancrofti or from animal filaria Setaria cervi? A description of protein production and a SDS-PAGE gel of the protein is needed.
- How much similarity exists in sequences of FSAg between W. bancrofti and S. cervi? At least a Western blot is needed to show antisera anti-W. bancrofti FSAg can cross-recognize S. cervi FSAg.
- Except for antibody passive transfer, is there any data for its immunogenicity and vaccine efficacy in vaccinated animal against challenge of microfilaria infection?
- Except for the therapeutic effect of anti-FSAg antisera against S. cervi infection, is there any prophylactic effect of the antisera to prevent S. cervi infection (administration before challenge)?
- Some minor errors
- Line 104: “Concentration of anti-FSAg antibodies in the serum was detected by ELISA”. Without purification, how did you measure the specific antibody (IgG?) concentration?
- Line 170 “anti-FSAg antigen” should be antibody
- Line 189 “1/6th of the initially injected Mf were present in the blood stage (patent) infection”. My understand the microfilaria appeared in the blood was not directly from challenged ones, should be released by the adult worms developed from challenged microfilaria.
- Since there is no purified antibody used in this experiment, all anti-FSAg antibodies should be changed to anti-FSAg antisera.

Author Response
Reviewer 1
Comments and Suggestions for Authors
Authors performed passive transfer of antisera against a microfilarial sheath antigen (FSAg) to rats infected with animal filaria Setaria cervi, to evaluate its efficacy of immunotherapy on filarial infection and filariasis. The results showed the significant reduction of microfilaria burden in blood and inhibition of pro-inflammatory cytokines that contribute to morbidity. Authors further predicted several B-cell and T-cell epitopes in FSAg using in-silico algorithm and predicted the construct from these epitopes may induce high titer of IgM and IgG1 in immunized host. The results provide further evidence FSAg could be a vaccine or immunotherapeutic target against LF. However, It means some questions and concerns needed to be addressed:
Authors’ Response
Authors express grateful thanks to learned reviewer for the valuable comments and propositions to improve the superiority of manuscript. We have tracked the recommendations provided by the reviewer and addressed them all in the revised manuscript.
1) How was FSAg protein produced? Is it a parasite-derived product or a recombinant protein? Is it from human filaria Wuchereria bancrofti or from animal filaria Setaria cervi? A description of protein production and a SDS-PAGE gel of the protein is needed.
Authors’ Response
Authors express their sincere thanks to the learned reviewer for reviewing our work and suggesting to modify the manuscript.
FSAg, is a filarial sheath protein, has been isolated and characterized from the surface of Wuchereria bancrofti as depicted in our previous reports [Mukherjee et al., 2017 & 2019].
It is a filarial parasite-derived protein that belongs to the nematode-bestrophin superfamily consisting of 399 amino acids [Mukherjee et al., 2017].
It is from the human filarial parasite Wuchereria bancrofti.
Regarding description of the protein production and SDS-PAGE, we request you to please go through our previous publication where we have already expressed all the technical details of protein extraction [Mukherjee et al., 2017].
We have briefly clarified these points in the revised text (lines 100-101).
Reference
Mukherjee, S.; Mukherjee, S.; Maiti, T.K.; Bhattacharya, S.; Sinha Babu, S.P. A novel ligand of toll-
like receptor 4 from the sheath of Wuchereria bancrofti microfilaria induces proinflammatory response
in macrophages. J. Infect. Dis. 2017, 215, 954–965, doi:10.1093/infdis/jix067.
Mukherjee, S.; Karnam, A.; Das, M.; Babu, S.P.S.; Bayry, J. Wuchereria bancrofti filaria activates human
dendritic cells and polarizes T helper 1 and regulatory T cells via toll-like receptor 4. Commun. Biol.
2019, 2, 169, doi:10.1038/s42003-019-0392-8.
2) How much similarity exists in sequences of FSAg between W. bancrofti and S. cervi? At least a Western blot is needed to show antisera anti-W. bancrofti FSAg can cross-recognize S. cervi FSAg.
Authors’ Response
Authors thanks to the learned reviewer for his/her valuable query regarding similarity between the FSAg of W. bancrofti and S. cervi. Yes, we have found very high degree of cross-reactivity between anti-W. bancrofti FSAg antisera and S. cervi FSAg (Mukherjee et al., 2020). This observation has already been published recently [Mukherjee et al., 2020].
We have included this information in the discussion part of the revised text (lines 359-360).
Reference
Mukherjee, S.; Joardar, N.; Sinha Babu, S.P. Exploring the homolog of a novel proinflammatory
microfilarial sheath protein (MfP) of Wuchereria bancrofti in the adult-stage bovine filarial parasite
Setaria cervi. J. Helminthol. 2020, 94, e15, doi:10.1017/S0022149X18001050
3) Except for antibody passive transfer, is there any data for its immunogenicity and vaccine efficacy in vaccinated animal against challenge of microfilaria infection?
Authors’ Response
Sincere thanks to the learned reviewer for this comment. However, we don’t have such kind data.
4) Except for the therapeutic effect of anti-FSAg antisera against S. cervi infection, is there any prophylactic effect of the antisera to prevent S. cervi infection (administration before challenge)?
Authors’ Response
Thanks for pointing out this issue. We have a data on this which suggests that anti-FSAg sera also possesses prophylactic effect to prevent S. cervi infection. Actually, our plan was to elaborate this part to incorporate in another manuscript. However, considering the query of the reviewer we have incorporated a part in the revised manuscript (Lines 240-245, Figure 2D & E in the revised manuscript).
5) Some minor errors
Line 104: “Concentration of anti-FSAg antibodies in the serum was detected by ELISA”. Without purification, how did you measure the specific antibody (IgG?) concentration?
Authors’ Response
Thanks for the comment. We have the purified antigen i.e. FSAg and using this antigen we performed an ELISA by coating the plates with FSAg (lines 107-108).
Line 170 “anti-FSAg antigen” should be antibody
Authors’ Response
Authors thanks to learned reviewer to pointing out the error and we have corrected the line in the revised manuscript.
Line 189 “1/6th of the initially injected Mf were present in the blood stage (patent) infection”. My understand the microfilaria appeared in the blood was not directly from challenged ones, should be released by the adult worms developed from challenged microfilaria.
Authors’ Response
We respect the comment. However, microfilaria appeared in the blood was directly from challenged ones. Development of adult stage parasite from microfilria injection within such a short period is seem to be impossible and has not been reported in any literature.
Since there is no purified antibody used in this experiment, all anti-FSAg antibodies should be changed to anti-FSAg antisera.
Authors’ Response
Thanks for the comment, we have changed all the ant-FSAg antibodies to anti-FSAg sera.
Reviewer 2 Report
This manuscript showed the efficacy of anti-bestrophin antibodies against experimental filariasis. Antibody treated animals displayed reduced parasite load, reduced proinflammatory cytokines. These new findings along with the previous studies of this group support that FSAg based antibody therapy could be a promising option to treat filarial infection and associated immunopathology. The experiment design and outcomes are sound and the manuscript is well written. I have a few comments listed below for the author's consideration to improve the manuscript.
1. Authors need to show representative smeared images collected to determine the Mf load.
2. As authors methodology showed that the antibodies are raised in the mouse model, hoe it is compatible to tests in rat model?
3. Authors need to show the data for immune simulation.
4. Typo in line 313.
Author Response
Reviewer 2
Comments and Suggestions for Authors
This manuscript showed the efficacy of anti-bestrophin antibodies against experimental filariasis. Antibody treated animals displayed reduced parasite load, reduced proinflammatory cytokines. These new findings along with the previous studies of this group support that FSAg based antibody therapy could be a promising option to treat filarial infection and associated immunopathology. The experiment design and outcomes are sound and the manuscript is well written. I have a few comments listed below for the author's consideration to improve the manuscript.
Authors’ Response
Authors express grateful thanks to learned reviewer for the valuable comments and suggestions to improve quality of the manuscript. We have followed the recommendations provided by reviewer and incorporated all in the revised manuscript.
- Authors need to show representative smeared images collected to determine the Mf load.
Authors’ Response
Authors are thankful to learned reviewer for the valuable suggestion and in the revised manuscript (Figure 2C & E), we have incorporated smeared images that determine mf load.
- As authors methodology showed that the antibodies are raised in the mouse model, how it is compatible to tests in rat model?
Authors’ Response
Thanks to the reviewer for this comment. Yes, anti-FSAg antisera was compatible in rat model. By adoptive transfer of anti-FSAg antisera, we aim to target the parasite antigen FSAg. Actually, anti-FSAg sera could also be induced in rat model but the amount of protein required for this is higher and the level of response is low as we repeatedly found in the past. On the other hand, anti-FSAg serum in mice could be raised with high yield with less protein antigen. Moreover, anti-FSAg obtained from mice does not produce any perturbation in the inflammatory homeostasis of rat (given in the text as well). Previous report has shown that immunoglobulin heavy chain locus of the rat has striking homology to mouse antibody genes (Proc Natl Acad Sci U S A. 1986 Aug; 83(16): 6075–6079).
Production of large amounts of recombinant FSAg in the near future and selection of an appropriate adjuvant would solve these issues and allow us to obtain rat anti-FSAg an-tisera for the experiments.
We have discussed these points in the discussion part of the revised text (Lines 367-375)
- Authors need to show the data for immune simulation.
Authors’ Response
Authors express grateful thanks to learned reviewer for his kind suggestion. We have used C-immSim server (http://150.146.2.1/C-IMMSIM/) to perform the immune simulation to predict immunogenicity and immune response. The algorithm is default and we have tested the efficacy of the refined FSAg using the default parameters applicable to study human immune response.
We have provided the information in Methods section of the revised text (Section 2.14, lines 188-193)
- Typo in line 313.
Authors’ Response
Author express sincere thanks to learned reviewer to pointing out the typing error. We have checked and corrected the typo in the revised manuscript.
Reviewer 3 Report
Minor grammar review may benefit this paper.
Author Response
Reviewer 3
Comments and Suggestions for Authors
Minor grammar review may benefit this paper.
Authors’ Response
Authors express grateful thanks to the learned reviewer for this comment. We have worked on the manuscript to enhance the language of the manuscript. We have checked the grammar manually with the help of an external expert.
Reviewer 4 Report
FSAg induced the secretion of cytokines of IL-6, IL-12 and IL-8 in addition to TNF-alpha and IL-1 beta. Whether was the effect on those profinflammatory cytokines investigated in this study? Besides that, the positive control was also recommended to include in the study. The concentration-dependent effecs of antibodies on the level of cytokines would result in more convincing conclusion for the effects on the inflammation.
I would recommend verifying the axis labels and legend in figure 2 and 3 for the FSAg antibodies rather than the FSAg.
Author Response
Reviewer 4
Comments and Suggestions for Authors
1) FSAg induced the secretion of cytokines of IL-6, IL-12 and IL-8 in addition to TNF-alpha and IL-1 beta. Whether was the effect on those profinflammatory cytokines investigated in this study? Besides that, the positive control was also recommended to include in the study. The concentration-dependent effects of antibodies on the level of cytokines would result in more convincing conclusion for the effects on the inflammation.
Authors’ Response
Authors express grateful thanks to learned reviewer for his kind suggestions. We have already published another work on FSAg / MfP and described details on the effect of proinflammatory cytokines [Mukherjee et al., 2019]. The effect of FSAg on those proinflammatory cytokines was not investigated in this study rather we worked on investigating the immunotherapeutic potential of anti-FSAg serum to reduce proinflammatory cytokines and ameliorate the infection level.
As recommended by the reviewer, we have incorporated the data obtained from the positive control (Figure 3, lines 255-257).
We have observed no significant reduction in the cytokine level at 50 µl/100 gm body weight, while treatment at 100 µl/100 gm body weight was found effective. In case of 200 µl/100 gm body weight dose, the effect was similar to that of 100 µl/100 gm body weight. So dose dependency was not observed (Lines 254-255).
Reference
Mukherjee, S.; Karnam, A.; Das, M.; Babu, S.P.S.; Bayry, J. Wuchereria bancrofti filaria activates human
dendritic cells and polarizes T helper 1 and regulatory T cells via toll-like receptor 4. Commun. Biol.
2019, 2, 169, doi:10.1038/s42003-019-0392-8.
2) I would recommend verifying the axis labels and legend in figure 2 and 3 for the FSAg antibodies rather than the FSAg.
Authors’ Response
Authors are thankful to the learned reviewer for pointing out these important errors, and we have corrected these sections of the figure 2 and 3 in revised manuscript.
Round 2
Reviewer 1 Report
Authors have revised the manuscript accordingly and the revised manuscript minor change required:
Line 358-9: "In addition, we have previously demonstrated very high degree of cross-reactivity between anti-W. bancrofti FSAg sera and S. cervi FSAg" should be : In addition, we have previously demonstrated that W. bancrofti FSAg and S. cervi FSAg shared high degree of antigenicity.
Line 124. "The concentration of anti-FSAg antibodies in the serum as measured by ELISA was 50.13±5.24 g/ml". As I said, ELISA can only measure the antibody titer, can not measure the concentration of antibody. If you want to measure the antibody concentration, you should purify the anti-FSAg IgG using antigen-affinity column, then measure the concentration of purified IgG using OD280 or BCA method
Author Response
Authors’ Response Comments and Suggestions for Authors
Authors have revised the manuscript accordingly and the revised manuscript minor change required:
1) Line 358-9: "In addition, we have previously demonstrated very high degree of cross-reactivity between anti-W. bancrofti FSAg sera and S. cervi FSAg" should be : In addition, we have previously demonstrated that W. bancrofti FSAg and S. cervi FSAg shared high degree of antigenicity.
Authors’ Response
Thank you for the suggestion and accordingly the sentence has been modified (lines 361-362)
2) Line 124. "The concentration of anti-FSAg antibodies in the serum as measured by ELISA was 50.13±5.24 mg/ml". As I said, ELISA can only measure the antibody titer, can not measure the concentration of antibody. If you want to measure the antibody concentration, you should purify the anti-FSAg IgG using antigen-affinity column, then measure the concentration of purified IgG using OD280 or BCA method.
Authors’ Response
Authors thank the learned reviewer for this valuable comment and the point has been addressed accordingly in the revised manuscript. We have now mentioned antibody titer instead of antibody concentration in the revised manuscript (lines 124-127).